

# Comment on the paper "Insignificant effect of climate change on winter haze pollution in Beijing" by Shen et al. (2018)

Run Liu[1], Lu Mao[2], Shaw Chen Liu[1*], Yuanhang Zhang[2], Hong Liao[3], Huopo Chen[4], Yuhang Wang[5]

[1]Institute for Environmental and Climate Research, Jinan University, Guangzhou 510632, China
[2]State Key Joint Laboratory of Environmental Simulation and Pollution Control, College of Environmental Sciences and Engineering, Peking University, Beijing 100871, China
[3]School of Environmental Science and Engineering, Nanjing University of Information Science & Technology, Nanjing, 210044, China
[4]Institute of Atmospheric Physics, Chinese Academy of Sciences, Beijing, 100029, China
[5]School of Earth and Atmospheric Sciences, Georgia Institute of Technology, Atlanta, GA 30332, USA

*Correspondence to*: Shaw Chen Liu (shawliu@jnu.edu.cn)

**Abstract.** The recent paper by Shen et al. (2018) (referred to hereafter as SHEN) made a sweeping statement on the winter haze pollution in Beijing by claiming "Insignificant effect of climate change on winter haze in Beijing". We argue that the paper contains three serious flaws. Either one of the three flaws can nullify the claim of SHEN.

SHEN made a sweeping statement on the winter haze pollution in Beijing by claiming "Insignificant effect of climate change on winter haze in Beijing". While failing to acknowledge the large differences in dataset used, analysis methodology, winter month selected, geographic region chosen, and period and time scale of study from the others, SHEN attempted to invalidate a number of previous studies, including Wang et al. (2015), Cai et al. (2017), Zou et al. (2017), and Li et al. (2018), which have suggested that climate change will worsen haze pollution in Beijing. In this context, our recent study (Mao et al., 2018)
also suggested that global warming and other climate changes such as El Nino-Southern Oscillation (ENSO) and Pacific Decadal Oscillation (PDO) contributed significantly to the trend as well as interannual variabilities of winter haze days in eastern China.

We have found three critical flaws in SHEN. First, SHEN did not make any evaluation of the accuracies or uncertainties of the projected changes in surface relative humidity (RH) and latitudinal wind velocity at 850 hPa (V850) in the RCP8.5 scenarios
calculated by an ensemble of 32 Coupled Model Intercomparison Project Phase 5 (CMIP5) climate models for the 21st century (2080-2099 vs. 2000-2019). Here we evaluate the accuracies and uncertainties of the projected changes in RH of CMIP5 climate models by comparing changes in RH and V850 from historical simulations (1960-2017) of these climate models to observed values. Figure 1a shows the values of linear trends of annual average RH in Beijing-Tianjin-Hebei (BTH) calculated for 1960-2017 historical simulations by an ensemble of 17 CMIP5 climate models (Table 1). A few models show significant
positive trends, but the average trend is only about 0.3% per decade. This small trend is consistent with the projected insignificant trends in 21st century (2080-2099 vs. 2000-2019) of RH in the RCP8.5 scenarios from an ensemble of 32 CMIP5



climate models as shown in Figure 5c of SHEN. In contrast, the small positive trend is in stark disagreement with the average trend of about -0.8% per decade observed at seven meteorological stations in BTH between 1973 and 2016 (Figure 1b). The disagreement is further illustrated in Figures 2a and 2b where the spatial distribution of trends of annual average RH in China calculated for 1960-2017 historical simulations by an ensemble of 17 CMIP5 climate models is compared to observed trends.

The model trends are positive in the north and negative in southern China, while observed trends are consistently negative and greater in values. These disagreements raise serious doubt on the validity of projected changes in RH in Beijing for the RCP8.5 scenarios by an ensemble of 32 CMIP5 climate models. This result is not surprising as the evaluation of climate models since IPCC AR5 assessed median-and-above model performance only for the projected global average temperature trends (Flato et al., 2013).

Second, Figure 1d of SHEN showed time series of monthly average $PM_{2.5}$ and three meteorological parameters, i.e. RH, V850, and PC1. The correlations among $PM_{2.5}$, RH, V850, and PC1 are very good as reported in SHEN. However, most of the good correlation is contributed by the large monthly variations. Will the good correlation hold true for yearly variations, and more importantly, hold for the time scale of climate change, which is the time scale of concern for SHEN? In addition, will the ratios between $PM_{2.5}$ and the three meteorological parameters of longer time scales remain the same as those derived from monthly

data? SHEN did not address these questions. Here we reproduce Figure 1d of SHEN in Figure 3a. Correlation coefficients of $PM_{2.5}$ with PC1, V850 and RH derived from Figure 3a are 0.90, 0.81 and 0.79 respectively, consistent with SHEN. In comparison, Figure 3b shows yearly average values of $PM_{2.5}$, PC1, V850 and RH; their corresponding correlation coefficients are 0.80, 0.66 and 0.46 respectively. These yearly values are significantly smaller than the monthly values, casting serious doubt on the applicability of results of monthly correlation to longer time scales. A further issue is that SHEN did not document

what parameters were used in the principal component analysis and how PC1 was derived.

Third, a more fundamental question is that a parameter such as PC1 should not be considered as a sole/exclusive/sufficient proxy of $PM_{2.5}$ just because PC1 has a good correlation with $PM_{2.5}$. In other words, PC1, V850 or RH should not be used to exclude other proxies such as those suggested by Wang et al. (2015), Cai et al. (2017), Zou et al. (2017), and Li et al. (2018). The exclusiveness (or sufficient condition) of an index can only be established if a mechanistic model that uses the index as a

sole proxy, can successfully reproduce the concentrations and trend of $PM_{2.5}$ quantitatively. SHEN did not develop such a model. For example, the variation of severe haze is associated with the daily variation of weather condition as shown in Cai et al. (2017) instead of the monthly PC1 given by SHEN. By using the same data as in SHEN, the correlation coefficient of PC1 with $PM_{2.5}$ on a daily basis is 0.68 (Figure 4b). Comparing to the monthly value of 0.90 in Figure 4a, it again demonstrates that different correlation coefficients are found at different time scales. Furthermore, the correlation coefficient of PC1 with

$PM_{2.5}$ for severe haze days (days with daily mean $PM_{2.5}$ concentration    150 µg m$^{-3}$, as defined in Cai et al. (2017) is a small value of 0.34 (Figure 4c). Therefore, it is inappropriate to use monthly PC1 to predict future severe winter haze pollution in Beijing as in SHEN. Compared to the large uncertainties in regional RH from the climate models in SHEN, haze weather index (HWI) in Cai et al. (2017) is defined by anomalies in large-scale circulation with a 3-dminetional dynamical concept, which can be captured by climate models for the past and future (see Cai et al. (2017) for the justification).





**Data availability**

The CMIP5 model results provided by World Climate Research Programme CMIP5 (http://cmip-pcmdi.llnl.gov/cmip5/, last access: 31 January 2019) are available. The data of this paper are available upon request to S. Liu (shawliu@jnu.edu.cn).

**Author contributions**

RL, LM, and SL performed most the analysis. RL, SL and HL prepared the manuscript with contributions from all coauthors.

**Competing interests**

The authors declare that they have no conflict of interest.

**Acknowledgements**

This work was supported by the Major Program of the National Natural Science Foundation of China [Grant number 91644222]
and the Environmental Public Welfare Industry in China [Grant number 201509001].

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



**Tables**

Table 1. Abbreviation and name of 17 CMIP5 models used in this study

| Abbreviation | Expanded model name |
| --- | --- |
| ACCESS1-0 | Commonwealth Scientific and Industrial Research Organisation Australian Community Climate and Earth System, version 1.0 |
| ACCESS1-3 | Commonwealth Scientific and Industrial Research Organisation Australian Community Climate and Earth System, version 1.3 |
| CNRM-CM5 | Centre National de Researches Météorologiques Coupled Global Climate Model, version 5 |
| CSIRO-Mk3-6-0 | Commonwealth Scientific and Industrial Research Organisation Mark, version 3.6.0 |
| CanESM2 | The second generation Canadian Earth System Model |
| FGOALS-S2 | The Flexible Global Ocean-Atmosphere-Land System model, Spectral Version 2 |
| HadGEM2-AO | Atmosphere and Ocean (non-Earth System version) configuration of HadGEM2 |
| HadGEM2-CC | Hadley Global Environment Model 2 - Carbon Cycle |
| HadGEM2-ES | Hadley Global Environment Model 2 - Earth System |
| INMCM4 | Institute of Numerical Mathematics Coupled Model, version 4.0 |
| MIROC-ESM-CHEM | An atmospheric chemistry coupled version of MIROC-ESM |
| MIROC-ESM | Model for Interdisciplinary Research on Climate Earth System Model |
| MIROC5 | Model for Interdisciplinary Research on Climate, version 5 |
| MRI-CGCM3 | Meteorological Research Institute Coupled Atmosphere–Ocean General Circulation Model, version 3 |
| MRI-ESM1 | Meteorological Research Institute-Earth System Model Version 1 |
| NorESM1-M | Norwegian Earth System Model, version 1, intermediate resolution |
| NorESM1-ME | Norwegian Climate Centre Earth System Model ME |



## Figures

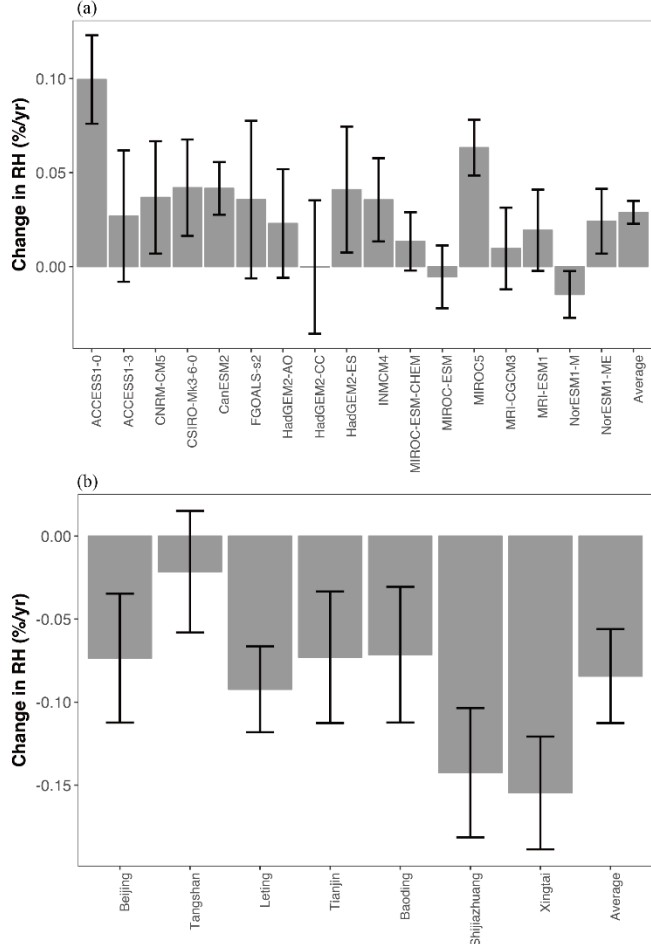

**Figure 1: (a) Linear trends of annual average RH (in % per year) in Beijing-Tianjin-Hebei (BTH) calculated for 1960-2017 historical simulations by an ensemble of 17 CMIP5 climate models. (b) Same as (a) except derived from NCDC station data.**





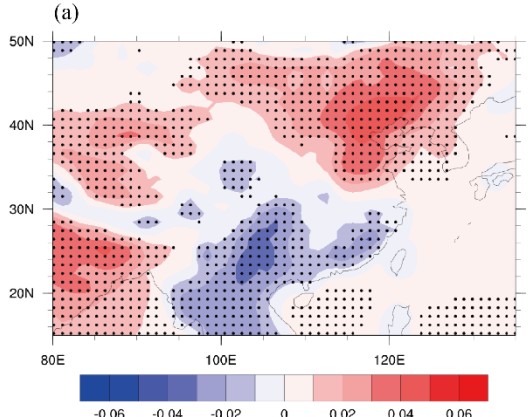

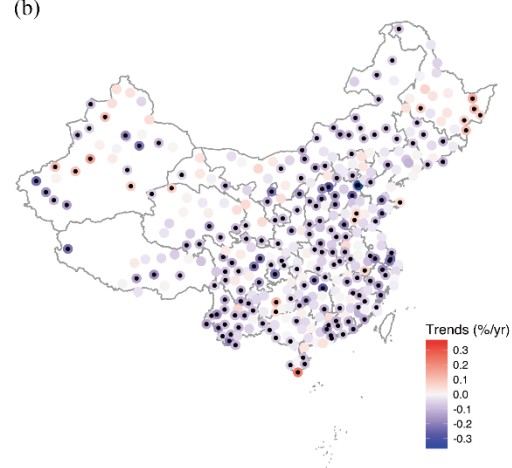

**Figure 2: (a) Spatial distribution of linear trends of annual average RH (in % per year) in China calculated for 1960-2017 historical simulations by an ensemble of 17 CMIP5 climate models. (b) Same as (a) except derived from NCDC station data.**



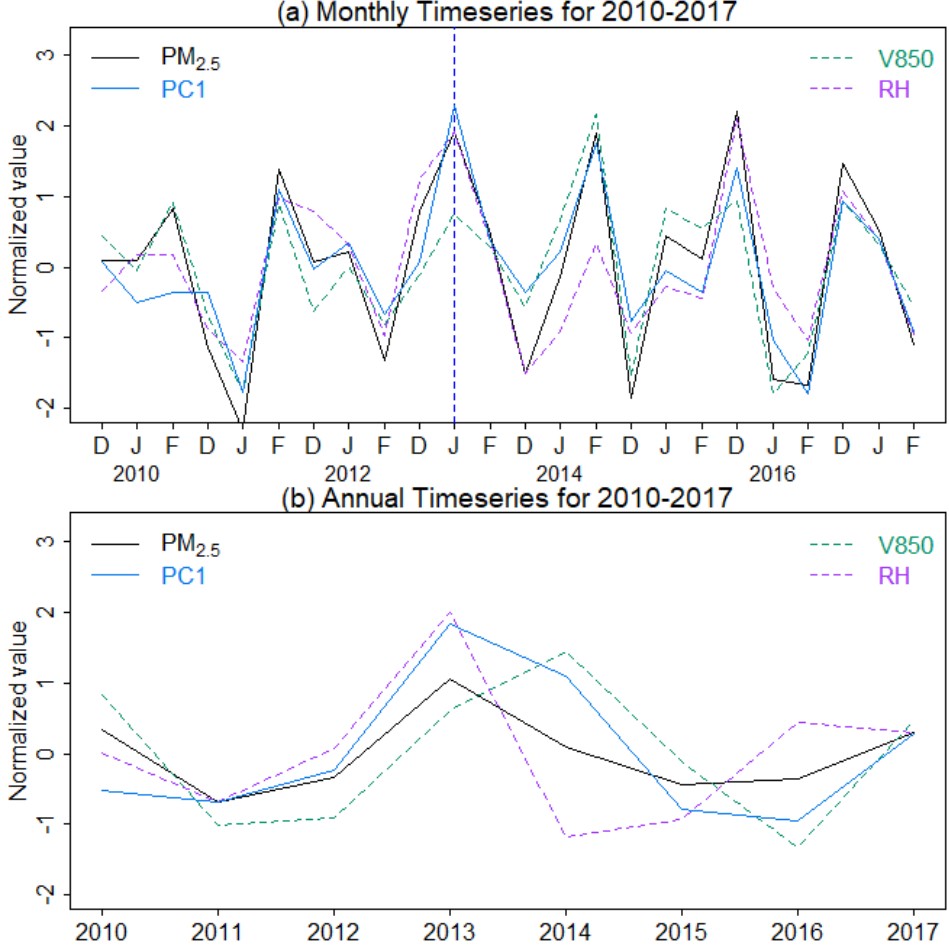

Figure 3: (a) Monthly mean time series for 2010–2017 of normalized PC1, PM$_{2.5}$, V850, and RH, the normalization is relative to the 2010–2017 means. (b) Same as (a) except for yearly means.



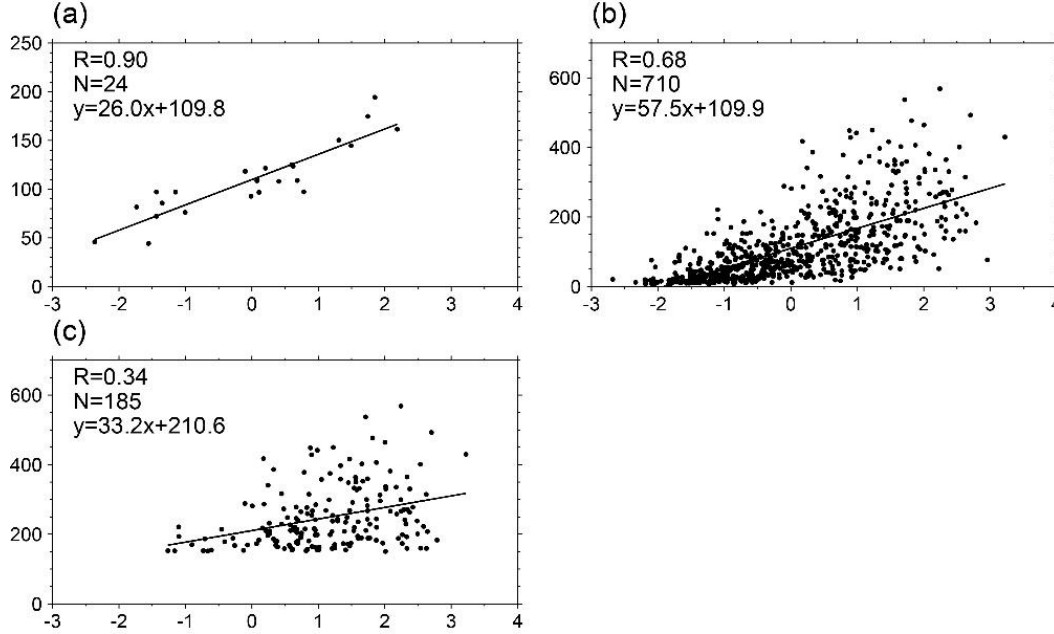

**Figure 4: Correlations between PC1 (defined by V850 and RH in SHEN, horizontal axis) with observed wintertime PM$_{2.5}$ concentrations in Beijing (µg m$^{-3}$, vertical axis) for (a) monthly PM$_{2.5}$ concentrations and PC1, (b) daily PM$_{2.5}$ concentrations and PC1, and (c) daily PM$_{2.5}$ concentrations and PC1 for severe haze days (daily mean PM$_{2.5}$ ≥ 150 µg m$^{-3}$). In each panel, N is the number of samples in the studied time period of 2010-2017 as in SHEN, and *R* is the correlation coefficient.**