# Peer review of "Comment on the paper "Insignificant effect of climate change on winter haze pollution in Beijing" by Shen et al. (2018)"

_Atmospheric Chemistry and Physics, 2019_

## Short Comment (SC1) · 17 Mar 2019

We thank Liu et al. (2019) for their careful reading of our work (Shen et al., 2018). Building on previous studies (Cai et al., 2017; Zou et al., 2017), Shen et al. (2018) found the effect of climate change on winter haze pollution in Beijing is uncertain and likely to be small. One difference with previous studies is that Shen et al. (2018) used relative humidity (RH) as a predictor meteorological variable, based on evidence that high RH is a driving variable for winter haze (Wang Y.S. et al., 2014; Wang Y.X. et al., 2014; Song et al., 2018). Put another way, we focused on the effects of climate change on Beijing haze from a different perspective from previous work.

Fig. 1-2 of Liu et al. (2019) display the trend of annual mean relative humidity (RH) during 1960-2017 from an ensemble of 17 CMIP5 models, arguing that the models cannot reproduce the observed trend. But more relevant to the winter haze problem is to examine RH trends in winter, as was done by Shen et al. (2018) for 1973-2017. There are a lot of missing data in the in-situ meteorological observations before 1973. Pendergrass et al. (2019) found in fact that the CMIP5 models could reproduce the observed wintertime trend of RH during 1973-2016 (see their Figure S3)

Shen et al. (2018) pointed out the strong correlation of wintertime $PM_{2.5}$ in Beijing with the first principal component PC1 of RH and the 850 hPa meridional wind velocity (V850), and subsequently used PC1 as a predictor variable for the effect of climate change on $PM_{2.5}$. Fig. 3 of Liu et al. (2019) shows that the interannual correlation ($r$) of PC1 and $PM_{2.5}$ is 0.80 during 2010-2017. Liu et al. concluded that this correlation is significantly smaller than the monthly one ($r = 0.90$) reported by Shen et al. (2018). The Liu et al. (2019) correlation is based on only 8 data points and hence less robust than the one derived by Shen et al. (2018), where individual months can be largely viewed as independent data points. In any case, a correlation coefficient of 0.80 as found by Liu et al. (2019) is still very high and a good basis for future-climate projections.

Fig. 4 of Liu et al. (2019) shows that the correlation of PC1 and $PM_{2.5}$ is only 0.34 when $PM_{2.5}$ concentrations are greater than 150 $\mu$g m$^{-3}$. But we doubt that the authors would be able to find high and robust correlations for such high $PM_{2.5}$ concentrations with any meteorological variables. This is because the high tail of a frequency distribution does not follow Gaussian statistics and reliance on the Pearson correlation coefficient is not recommended. Extreme value theory is a better way to characterize the distribution of high $PM_{2.5}$ in relation to meteorological variables, as presented by Pendergrass et al. (2019) for Beijing winter haze. Indeed, among a variety of meteorological variables, Pendergrass et al. (2019) found that the best fit to the high tail of the $PM_{2.5}$ distribution in a Poisson point process model is RH and V850. Pendergrass et al. (2019) went on to apply their point process model to future climate projections and found no increase in extreme winter haze frequency, consistent with Shen et al. (2018). They found that the only alternative model projecting an increase in haze frequency was one that did not include RH as predictor variable, but that model performed very poorly in the Aikake test.

Liu et al (2019) argue that the PC1 should not be used to exclude other proxies used in previous studies. In fact, Shen et al. (2018) and Pendergrass et al (2019) did consider all the meteorological variables used in Cai et al. (2017) and Zou et al. (2017). See Table S1 and Figures S3-S4 in Shen et al. (2018),

and Figures 4 and S1-S2 and related discussion in Pendergrass et al. (2019). Shen et al. (2018) consistently found that these additional variables produced a poorer statistical model than RH and V850 alone. Shen et al. (2018) also discussed why we did not ultimately use zonal wind at 500 hPa (U500) and the temperature gradient between 850 and 250 hPa to infer future trends of $PM_{2.5}$ (Figure S11 and Section 5).

Finally, we thank Liu et al. (2019) for drawing attention to the uncertainty in RH trends. Climate models project a general decrease of RH across eastern China in the future (Byrne and O'Gorman, 2013; Lau and Kim, 2015), but we acknowledge that the uncertainty could be large on a smaller, regional scale as in Beijing.

**Reference**

Byrne, M. P., and O'Gorman, P.A.: Link between land-ocean warming contrast and surface relative humidities in simulations with coupled climate models, Geophys. Res. Lett., 40, doi:10.1002/grl.50971, 2013.

Lau, W.K.M., and Kim, K. M.: Robust Hadley Circulation changes and increasing global dryness due to CO2 warming from CMIP5 model projections, Proc. Natl. Acad. Sci. USA, 112, 3630–3653, 2015.

Cai, W., Li, K., Liao, H., Wang, H., and Wu, L.: Weather conditions conducive to Beijing severe haze more frequent under climate change, Nat. Clim. Change, 7, 257–262, https://doi.org/10.1038/nclimate3249, 2017.

Pendergrass, D. C., Shen, L., Jacob, D. J., & Mickley, L. J. (2019). Predicting the impact of climate change on severe wintertime particulate pollution events in Beijing using extreme value theory. Geophysical Research Letters, 46, 1824–1830. https://doi.org/10.1029/ 2018GL080102.

Song, S., Gao, M., Xu, W., Sun, Y., Worsnop, D. R., Jayne, J. T., Zhang, Y., Zhu, L., Li, M., Zhou, Z., Cheng, C., Lv, Y., Wang, Y., Peng, W., Xu, X., Lin, N., Wang, Y., Wang, S., Munger, J. W., Jacob, D. J., and McElroy, M. B.: Possible heterogeneous chemistry of hydroxymethanesulfonate (HMS) in northern China winter haze, Atmos. Chem. Phys., 19, 1357-1371, https://doi.org/10.5194/acp-19-1357-2019, 2019.

Wang, Y.S., Yao, L., Wang, L., Liu, Z., Ji, D., Tang, G., Zhang, J., Sun, Y., Hu, B., and Xin, J.: Mechanism for the formation of the January 2013 heavy haze pollution episode over central and eastern China, Sci. China Earth Sci., 57, 14–25, 2014

Wang, Y.X., Zhang, Q., Jiang, J., Zhou, W., Wang, B., He, K., Duan, F., Zhang, Q., Philip, S., and Xie, Y.: Enhanced sulfate formation during China's severe winter haze episode in Jan 2013 missing from current models, J. Geophys. Res.-Atmos., 119, 10425-10440, doi:10.1002/2013JD021426, 2014.

Zou, Y., Wang, Y., Zhang, Y., and Koo, J. H.: Arctic sea ice, Eurasia snow, and extreme winter haze in China, Sci. Adv., 3, e1602751, 2017.

---

## Author Comment (AC1) · 21 Mar 2019

Shen (referred hereafter as Shen2019) made some interesting statements on the comments by Liu et al. (2019) on the paper Shen et al. (2018).

Liu et al. (2019) showed that the trend of annual mean relative humidity (RH) in China during 1960-2017 from an ensemble of 17 CMIP5 models cannot reproduce the observed trend. However, Shen2019 claimed that the CMIP5 models could reproduce the observed wintertime trend of RH during 1973-2016 in Beijing. We like to point out that climate models like those used in CMIP5 can make more reliable projections of trends over large regions and long-time scales than wintertime in Beijing. Furthermore, we have reproduced below Figures 1 and 2 of Liu et al. (2019) for wintertime, and found no agreement between the trend of wintertime RH in the Beijing-Tianjin-Hebei (BTH) region during 1960-2017 from an ensemble of 17 CMIP5 models and the observed trend (Figures 1 and 2). Therefore, we reaffirm the claim by Liu et al. (2019) that the ensemble of 17 CMIP5 models shows little skill in simulating the observed trend of annual or wintertime mean RH in BTH during 1960-2017.

Shen2019 claimed that a correlation coefficient of 0.80 of PC1 with $PM_{2.5}$ on a yearly basis as found by Liu et al. (2019) is still very high and a good basis for future-climate projections. We call the attention to the fact that the correlation coefficient of PC1 with $PM_{2.5}$ on daily basis is 0.68 (more data points than monthly values) as also found by Liu et al. (2019), which is not very high and much lower than the value of 0.90 used in the original paper. More importantly, Liu et al. (2019) pointed out that the correlation coefficient of PC1 with $PM_{2.5}$ changes with the time scale of interest, it may not stay high for the time scale of climate change, which is the time scale of concern for Shen et al. (2018). Shen2019 did not address this point.

In regard to the point that Liu et al. (2019) argued that the PC1 should not be used to exclude other proxies used in previous studies, Shen2019 implied that the extreme value analysis in a later study supported the original paper. The mathematical basis of that study is sufficiently different from the original paper to land credible support to neither the original paper nor the rebuttal of Liu et al. (2019). Whether the extreme value theory was applied appropriately in that study is also debatable. Furthermore, Shen2019 did not address the fundamental issue raised by Liu et al. (2019) that a parameter such as PC1 should not be considered as a sole/exclusive/sufficient proxy of $PM_{2.5}$ just because PC1 has a good correlation with $PM_{2.5}$.

In conclusion, we reaffirm the three critical flaws found in Shen et al. (2018).

**References**

Liu, R., Mao, L., Liu, S. C., Zhang, Y., Liao, H., Chen, H., and Wang, Y.: Comment on the paper Insignificant effect of climate change on winter haze pollution in Beijing by Shen et al. (2018), Atmos. Chem. Phys. Discuss., https://doi.org/10.5194/acp-2019-193, in review, 2019.

Shen, L., Jacob, D. J., Mickely, L. J., Wang, Y., and Zhang, Q.: Insignificant effect of climate change on winter haze pollution in Beijing, Atmos. Chem. Phys., 18, 17489–17496, https://doi.org/10.5194/acp-18-17489-2018, 2018.

**Figures**

[Figure]

Figure 1: (a) Linear trends of wintertime average RH (in % per year) in Beijing-Tianjin-Hebei (BTH) calculated for 1960-2017 historical simulations by an ensemble of 17 CMIP5 climate models. (b) Same as (a) except derived from NCDC station data.

[Figure]

[Figure]

**Figure 2: (a) Spatial distribution of linear trends of winter average RH (in % per year) in China calculated for 1960-2017 historical simulations by an ensemble of 17 CMIP5 climate models. (b) Same as (a) except derived from NCDC station data.**

---

## Short Comment (SC2) · 23 Mar 2019

We thank Liu et al. (2019) for the discussion. After careful reading of their objections to the Shen et al. (2018) work, we think that the Shen et al. (2018) work is correct and we do not agree with the Liu et al. (2019) objections.

Based on Figure S3 of Pendergrass et al. (2018), CMIP5 models can in general capture the wintertime RH trends during 1973-2016. We acknowledge that the uncertainty could be large on a smaller, regional scale as in Beijing. We do not include data prior to 1973 due to the large amount of missing data in insitu meteorological observations. This is the reason that the HadISDH dataset (Smith et al., 2011; Willett et al., 2014) only provides data after 1973 (https://www.metoffice.gov.uk/hadobs/hadisdh/). Now we re-plot the Figure S3 of Pendergrass et al. (2018) but with a larger ensemble of 26 CMIP5 models, and we find that our conclusion still holds. We also plot the trends for 1960-2017 in CMIP5 models, and the spatial pattern does not change.

 $PM_{2.5}$  concentrations are noisier on the daily scale, so it is not surprising to find that the daily correlation with meteorology is smaller than the monthly one. Also, the daily correlation of  $PM_{2.5}$  with the haze weather index used by Cai et al. (2017) is 0.60, even smaller than the correlation Shen et al. (2018) found with PC1 (r = 0.68).

Liu et al (2019) still argues that the use of PC1 should not exclude the use of other proxies. But in fact in our last reply, we made clear that Shen et al. (2018) considered all variables used in Cai et al. (2017) and Zou et al. (2017). Shen et al. (2018) also discussed why we did not ultimately use zonal wind at 500 hPa (U500) and the temperature gradient between 850 and 250 hPa to infer future trends of  $PM_{2.5}$  (Figure S11 and Section 5).

The extreme value model is more suitable for studying extreme events, as is well accepted in many previous studies (Coles, 2001; Rieder et al., 2013; Shen et al., 2016). The study of Pendergrass et al. (2019) was peer-reviewed, so in our view Liu et al. should give detailed reasons if they think this work is debatable.

**Reference**

Coles, S. G.: An Introduction to Statistical Modeling of Extreme Values, Springer, New York, 2001.

- Liu, R., Mao, L., Liu, S. C., Zhang, Y., Liao, H., Chen, H., and Wang, Y.: Comment on the paper Insignificant effect of climate change on winter haze pollution in Beijing by Shen et al. (2018), Atmos. Chem. Phys. Discuss., https://doi.org/10.5194/acp-2019-193, in review, 2019.
- Pendergrass, D. C., Shen, L., Jacob, D.J., and Mickley, L. J.: Predicting theimpact of climate change on severewintertime particulate pollution events in Beijing using extreme value theory, Geophys. Res. Lett., 46,1824–1830. https://doi.org/10.1029/2018GL080102, 2019.

- Rieder, H. E., Fiore, A. M., Polvani, L. M., Lamarque, J. F., and Fang, Y.: Changes in the frequency and return level of high ozone pollution events over the eastern United States following emission controls, Environ. Res. Lett., 8, 014012, doi:10.1088/1748-9326/8/1/014012, 2013.
- Shen, L., Mickley, L. J., Gilleland E. : Impact of increasing heat waveson U.S. ozone episodes in the 2050s: Results from a multimodel analysisusing extreme value theory, Geophys.Res. Lett.,43, 4017–4025, doi:10.1002/2016GL068432, 2016
- Shen, L., Jacob, D. J., Mickley, L. J., Wang, Y., and Zhang, Q.: Insignificant effect of climate change on winter haze pollution in Beijing, Atmos. Chem. Phys., 18, 17489-17496, https://doi.org/10.5194/acp-18-17489-2018, 2018.
- Smith, A., Lott, N., and Vose, R.: The Integrated Surface Database Recent Developments and Partnerships, B. Am. Meteorol. Soc., 92, 704–708, 2011.
- Willett, K. M., Dunn, R. J. H., Thorne, P. W., Bell, S., de Podesta, M., Parker, D. E., Jones, P. D., and Williams Jr., C. N.: HadISDH land surface multi-variable humidity and temperature record for climate monitoring, Clim. Past, 10, 1983–2006, doi:10.5194/cp10-1983-2014, 2014

**Wintertime RH trends in observations and CMIP5 models**

**Figure 1.** (a) Linear trends in observed RH from 1973 to 2016, calculated from the gridded monthly mean RH dataset HadISDH. This data is produced by the Hadley Centre for Climate Prediction and Research and is available online (https://www.metoffice.gov.uk/hadobs/hadisdh/). (b) The average of linear trends in CMIP5 simulations of ground-level RH over China in the same time period. Historical simulations are used for 1973-2005 and RCP4.5 simulations are used for 2006-2016. This panel is same as Figure S3 of Pendergrass et al. (2018) except it uses a larger ensemble of 26 CMIP5 models here. These models are ACCESS1-0, ACCESS1-3, BCC-CSM1-1, BNU-ESM, CCSM4, CNRM-CM5, CSIRO-Mk3, CanESM2, GFDL-CM3, GFDL-ESM2G, GFDL-ESM2M, GISS-E2-H, GISS-E2-R, HadCM3, HadGEM2-CC, HadGEM2-ES, INM-CM4, IPSL-CM5A-LR, IPSL-CM5A-MR, IPSL-CM5B-LR, MIROC-ESM-CHEM, MIROC-ESM, MIROC5, MRI-CGCM3, NorESM1-ME, NorESM1-M. (c) Same as (b) but for 1960-2017.

**3**

---

## Author Comment (AC2) · 11 Apr 2019

Shen (referred hereafter as Shen2019-2) made four additional short comments on our earlier response in this discussion series. With the following short responses to the four comments, we hope the discussion series can converge to an agreement not-to-agree and let the audience draw their own conclusions.

1. We believe the audience would agree that Figure 1 of Shen2019-2 does not support their statement- "CMIP5 models can in general capture the wintertime RH trends during 1973-2016", contrary to the claim of Shen2019-2.

2. Shen2019-2 agrees with our earlier statement- "the correlation coefficient of PC1 with $PM_{2.5}$ changes with the of time scale of interest". So they should agree that the correlation coefficient "may not stay high for the time scale of climate change".

3. Shen2019-2 still had not addressed the fundamental issue raised by Liu et al. (2019)- "that a parameter such as PC1 should not be considered as a sole/exclusive/sufficient proxy of $PM_{2.5}$ just because PC1 has a good correlation with $PM_{2.5}$". In fact, it is well-known that even a perfect correlation coefficient (1.0) does not imply any causal relationship, let alone an exclusive/sufficient relationship. Therefore, the sweeping claim by Shen et al. (2018) that "insignificant effect of climate change on winter haze in Beijing" is invalid.

4. The extreme value (EV) model is irrelevant to the discussion series because EV statistics addresses different questions from general correlation statistics. Having said that, the fundamental assumption of a Poisson process is that the EV events are independent from one another, which is not the case for haze events. For example, the high $PM_{2.5}$ episode in January 2013 lasted weeks. The EV events were obviously autocorrelated in this episode. For that reason, haze (or heat wave for that matter) EV events cannot be directly modelled as a Poisson process. Removing the autocorrelations in EV events will greatly reduce the size of the dataset. As such, it calls into question the validity of the study results in which the autocorrelation of EV data was not properly treated.

**References**

Liu, R., Mao, L., Liu, S. C., Zhang, Y., Liao, H., Chen, H., and Wang, Y.: Comment on the paper Insignificant effect of climate change on winter haze pollution in Beijing by Shen et al. (2018), Atmos. Chem. Phys. Discuss., https://doi.org/10.5194/acp-2019-193, in review, 2019.

Shen, L., Jacob, D. J., Mickely, L. J., Wang, Y., and Zhang, Q.: Insignificant effect of climate change on winter haze pollution in Beijing, Atmos. Chem. Phys., 18, 17489–17496, https://doi.org/10.5194/acp-18-17489-2018, 2018.

---

## Short Comment (SC3) · 22 Apr 2019

Comments: This paper entitled "Insignificant effect of climate change on winter haze pollution in Beijing" by Shen et al. used the first principal component (PC1) of V850 and RH as a haze proxy to investigate the effect of climate change on winter haze in Beijing. The relationship between PC1 proxy and the dipole structure of the Arctic sea ice and the Pacific SSTs is well documented. However, using the PC1 of V850 and RH as a haze proxy is arbitrary, thus derived conclusion claiming insignificant effect of climate change on winter haze pollution in Beijing is also questionable. The major comments are as follows:

[Figure]

1. 2010-2017 is a relatively stable period of pollution emissions. In this case, the role of meteorological factors is indeed relatively large, so the PC1 can well characterize the change of PM2.5. But what if there is an increasing trend in pollution emissions? Considering the limited length of PM2.5 data, it is suggested that haze days and visibility data should be compared with the PM2.5 and PC1 series, in order to determine whether the PC1 could be served as a proxy for haze under any pollution emission conditions.

2. In fact, climate change includes both human impacts (anthropogenic effect) and natural variability. It is unclear what the "climate change" in this paper refers to. The PC1 is a combination of V850 and RH anomalies. Their changes are related to PM2.5 during the period of 2010-2017, which is no problem. But their changes may also be related to human activities. A definite conclusion could be obtained only after the attribution analysis is carried out. However, there is no attribution analysis in this paper at present.

3. Some studies show that the increase of haze in Beijing was related to the weakening of winter monsoon over East Asia. Additionally, the Pacific and Atlantic SST showed interdecadal variations which are mainly AMO and PDO, which is mentioned in the paper. However, besides, the SST changes also include the climate warming trend and ENSO events. If the authors want to draw a conclusion on the effects of climate change on winter haze, the attribution analysis is needed to prove that the impact of human activities is insignificant or indirect on the physically-solid basis.

4. The paper claimed that in the future the PC1 had no significant trend under RCP8.5 scenario. We recommend that the trend analysis of PM2.5 simulated by ACCMIP (Atmospheric Chemistry and Climate Model Intercomparison Projection) should be carried out in order to see the future changes in PM2.5.

---

## Referee Comment (RC1) · Anonymous Referee #1 · 24 Apr 2019

This comment paper raises doubt on the recent paper by Shen et al. (2018), which draws the conclusion that the effect of climate change on winter haze in Beijing is small and uncertain. The authors point out three issues with Shen et al. (2018), which I think are reasonable arguments. The authors well addressed the questions posed by the Shen. The thoughtful debate on this controversial topic is worth publication at ACP, though I don't think the comment alone could nullify the conclusions of Shen et al. (2018). I have a few comments:

1. A major disagreement between Liu and Shen is whether CMIP5 models can capture the observed trend of RH. Given the large inter-annual variability of RH, the derived

trend may differ with the starting year. Shen points out that there are a lot of missing data in meteorological stations before 1973. Missing values could potentially lead to sampling biases and therefore biases in the trend. Shen argues that CMIP5 models can reproduce the trend between 1973 and 2016. To address this argument, I'd suggest the authors calculate the trends in RH between 1973 and 2016, and evaluate if CMIP5 models can capture the observed trend.

2. The second argument raised by the authors is that the good correlation between PC1 and PM2.5 derived from monthly data may not hold for other time scales. While I agree the correlation may vary with time scales, I don't think this analysis could really nullify the predictability of PC1. The correlation coefficient for annual mean is based on only eight data points, which is likely to be unstable. Qualitatively speaking, I could tell PC1 can capture most if not all the inter-annual variability of PM2.5. I tend to disagree with the statement that the yearly values are 'significantly smaller' than monthly values.

3. The authors pointed out that PC1 should not be used as a single proxy for PM2.5. Admittedly, a statistical proxy has uncertainties, but I don't think it's realistic to have a proxy that could perfectly simulate all the observed temporal variabilities. Shen et al. (2018) explain that their results differ from Cai et al. (2017) because Cai et al. (2017) does not include RH as a predictor, but such difference is not discussed in the comment. The different conclusions drawn from Shen et al. (2018), Cai et al. (2017) and Pendergrass et al. (2019) actually reflect the effect of climate change is uncertain and controversial. I don't think the conclusions of Shen et al. (2018) would be invalid just because of the inherent uncertainties of the chosen proxy.

---

## Author Response (AR1)

**Anonymous Referee #1:** This comment paper raises doubt on the recent paper by Shen et al. (2018), which draws the conclusion that the effect of climate change on winter haze in Beijing is small and uncertain. The authors point out three issues with Shen et al. (2018), which I think are reasonable arguments. The authors well addressed the questions posed by the Shen. The thoughtful debate on this controversial topic is worth publication at ACP, though I don't think the comment alone could nullify the conclusions of Shen et al. (2018). I have a few comments:

**Response:** We thank this referee for her/his perceptive comments. We appreciate that the referee found "the thoughtful debate on this controversial topic is worth publication at ACP." We have carefully considered all comments. Listed below are our point-by-point responses to individual comments (Referee's points in black, our responses in blue).

1. A major disagreement between Liu and Shen is whether CMIP5 models can capture the observed trend of RH. Given the large inter-annual variability of RH, the derived trend may differ with the starting year. Shen points out that there are a lot of missing data in meteorological stations before 1973. Missing values could potentially lead to sampling biases and therefore biases in the trend. Shen argues that CMIP5 models can reproduce the trend between 1973 and 2016. To address this argument, I'd suggest the authors calculate the trends in RH between 1973 and 2016, and evaluate if CMIP5 models can capture the observed trend.

**Response:** We agree that the key to this argument is that whether CMIP5 models can capture the observed trend of RH. Here, we calculate the trends in RH between 1960 and 2017 with data from China Meteorological Administration (CMA) rather than NCDC used in our original comment (acp-2019-193). The RH data from CMA, unlike data from NCDC, do not have the problem of missing values before 1973. The comparison is shown in Figures 1 and 2, which are in good agreement with the original figures. So, we have replaced those figures in the revised manuscript, but left the text essentially intact.

[Figure]

Figure 1: (a) Linear trends of wintertime average RH (in % per year) in Beijing-Tianjin-Hebei (BTH) calculated for 1960-2017 historical simulations by an ensemble of 17 CMIP5 climate models. (b) Same as (a) except derived from 25 meteorological stations of CMA in BTH region.

[Figure]

[Figure]

Figure 2: (a) Spatial distribution of linear trends of winter average RH (in % per year) in China calculated for 1960-2017 historical simulations by an ensemble of 17 CMIP5 climate models. (b) Same as (a) except derived from NCDC station data. Small black dots denote those trends significant at 95% confidence level.

2. The second argument raised by the authors is that the good correlation between PC1 and PM2.5 derived from monthly data may not hold for other time scales. While I agree the correlation may vary with time scales, I don't think this analysis could really nullify the predictability of PC1. The correlation coefficient for annual mean is based on only eight data points, which is likely to be unstable. Qualitatively speaking, I could tell PC1 can capture most if not all the inter-annual variability of PM2.5.

5 I tend to disagree with the statement that the yearly values are 'significantly smaller' than monthly values.

**Response:** In our first reply to the interactive comments, we pointed out the fact that, in addition to annual data, "the correlation coefficient of PC1 with $PM_{2.5}$ on daily basis (more data points than monthly values) is 0.68, which is significantly lower than the value of 0.9 used in the original paper." We made this point in the third section of the original manuscript. Therefore, the burden is on Shen and co-authors to prove that the correlation coefficient of PC1 with $PM_{2.5}$ stays high for the time scale of

10 climate change, which is the time scale of concern for Shen et al. (2018).

3. The authors pointed out that PC1 should not be used as a single proxy for PM2.5. Admittedly, a statistical proxy has uncertainties, but I don't think it's realistic to have a proxy that could perfectly simulate all the observed temporal variabilities. Shen et al. (2018) explain that their results differ from Cai et al. (2017) because Cai et al. (2017) does not include RH as a predictor, but such difference is not discussed in the comment. The different conclusions drawn from Shen et al. (2018), Cai

15 et al. (2017) and Pendergrass et al. (2019) actually reflect the effect of climate change is uncertain and controversial. I don't think the conclusions of Shen et al. (2018) would be invalid just because of the inherent uncertainties of the chosen proxy.

**Response:** We agree with the referee that the conclusions of Shen et al. (2018) would not be invalid just because of the inherent uncertainties of the chosen proxy. However, the conclusions of Shen et al. (2018) are invalid because of a fundamental point raised in our original manuscript: "
[revised manuscript text omitted]